# 1 The SPectrometer for Ice Nuclei (SPIN): An instrument to

## 2 investigate ice nucleation

- 3 S. Garimella<sup>1</sup>, T. B. Kristensen<sup>2</sup>, K. Ignatius<sup>2</sup>, A. Welti<sup>2</sup>, J. Voigtländer<sup>2</sup>, G. R.
- 4 Kulkarni<sup>3</sup>, F. Sagan<sup>4</sup>, G. L. Kok<sup>4</sup>, J. Dorsey<sup>5</sup>, L. Nichman<sup>5</sup>, D. Rothenberg<sup>1</sup>, M.
- 5 Rösch<sup>1</sup>, A. Kirchgäßner<sup>6</sup>, R. Ladkin<sup>6</sup>, H. Wex<sup>2</sup>, T. W. Wilson<sup>7</sup>, L. A. Ladino<sup>8</sup>, J. P. D.
- 6 Abbatt<sup>8</sup>, O. Stetzer<sup>9</sup>, U. Lohmann<sup>10</sup>, F. Stratmann<sup>2</sup>, and D. J. Cziczo<sup>1</sup>
- 7 [1] {Massachusetts Institute of Technology, Cambridge, MA, United States}
- 8 [2] {Leibniz Institute for Tropospheric Research, Leipzig, Germany}
- 9 [3] {Pacific Northwest National Laboratory, Richland, WA, United States}
- 10 [4] {Droplet Measurement Technologies, Boulder, CO, United States}
- 11 [5] {University of Manchester, Manchester, United Kingdom}
- 12 [6] {British Antarctic Survey, Cambridge, United Kingdom}
- 13 [7] {University of Leeds, Leeds, United Kingdom}
- 14 [8] {University of Toronto, Toronto, Canada}
- 15 [9] {V-ZUG AG, Zurich, Switzerland}
- 16 [10] {Swiss Federal Institute of Technology, Zurich, Switzerland}
- 17 Correspondence to: D. J. Cziczo (djcziczo@mit.edu)

## 1 Abstract

2 The SPectrometer for Ice Nuclei (SPIN) is a commercially available ice nuclei counter 3 manufactured by Droplet Measurement Technologies in Boulder, CO. The SPIN is a continuous 4 flow diffusion chamber with parallel plate geometry based on the Zurich Ice Nucleation 5 Chamber and the Portable Ice Nucleation Chamber. This study characterizes and describes the 6 behavior of the SPIN chamber, reports data from laboratory measurements, and quantifies 7 uncertainties associated with the measurements. A machine learning approach for analyzing 8 depolarization data from the SPIN Optical Particle Counter is also presented. Experiments with 9 ammonium sulfate are used to investigate homogeneous freezing and droplet breakthrough, 10 experiments with kaolinite, NX illite, and silver iodide are used to investigate heterogeneous ice 11 nucleation, and results are compared to those from the literature. Overall, we report that the SPIN 12 is able to reproduce previous CFDC ice nucleation measurements. 13

### 1 1. Introduction

2 Aerosol particles are required for the nucleation of cloud droplets and ice crystals in Earth's 3 atmosphere (Pruppacher and Klett, 1997). Ice nucleating particles (INP) facilitate the formation 4 of ice crystals via several possible mechanisms, including deposition nucleation, immersion 5 freezing, and contact freezing (Rogers, 1989; Pruppacher and Klett 1997). Though droplets freeze homogeneously below temperatures of ~-38°C, INP also facilitate freezing below water 6 7 saturation at such cold temperatures (Koop et al., 2000). Because of the complexity of the ice 8 nucleation process, understanding INP interactions with water has been difficult (Boucher et al., 9 2013; Stocker et al., 2013). Despite this difficulty, the significant influence that mixed-phase, 10 and ice clouds have on the Earth's radiative budget and hydrologic cycle makes understanding 11 the microphysics of cloud formation an important step in quantifying their influence on climate 12 (Boucher et al., 2013; Stocker et al., 2013).

13 Laboratory measurements allow for the investigation of ice nucleation at specific conditions with controlled aerosol properties and provide insight into ice formation as it occurs in the 14 15 atmosphere. Several types of laboratory instruments have been developed to measure the 16 efficiency of heterogeneous nucleation of cloud droplets and ice crystals, and many of these instruments have been deployed to conduct field observations. Among these instruments, the 17 Continuous Flow Diffusion Chamber (CFDC) (Rogers, 1988) has proven a useful tool to 18 19 measure the conditions required to nucleate ice crystals on various INP's. Studies have been 20 conducted on different nucleation and freezing mechanisms using many types of aerosol particles 21 under a wide range of temperatures and RH's (Rogers, 1988; Salam et al., 2006; Stetzer et al., 22 2008). Improved versions of the original cylindrical chamber described by Rogers (1980) have been successfully deployed in ground and aircraft based field campaigns (Chen et al., 1998; 23

DeMott et al., 2003a; DeMott et al., 2003b). Development of parallel plate chamber geometry 1 2 has simplified several technical aspects of the chamber design (e.g., lower chamber weight, less 3 complex machining, and simpler refrigeration plumbing than for the cylindrical geometry) at the 4 expense of edge effects (i.e., deviations from ideality at the chamber edges). One contemporary 5 parallel plate design is the Zurich Ice Nucleation Chamber (ZINC) (Stetzer et al., 2008), which has been used for several laboratory studies (e.g., Welti et al., 2009; Welti et al., 2014). The 6 7 Portable Ice Nucleation Chamber (PINC), designed as a field-deployable version of the ZINC, 8 has since been used to conduct several laboratory and field studies (Chou et al., 2011; Chou et 9 al., 2013; Kanji et al., 2013). In addition, other research groups have also developed similar 10 chambers (Kanji et al., 2009; Kulkarni et al., 2009; Friedman et al., 2011; Saito et al., 2011). 11 Adapting the parallel plate design and other features from the ZINC and PINC chambers, the 12 SPectrometer for Ice Nuclei (SPIN) is a commercially available ice nuclei counter manufactured 13 by Droplet Measurement Technologies in Boulder, CO. This study characterizes the behavior of 14 the SPIN chamber and reports data that validate the general design and performance.

15

#### 16 **2. Instrument theory and design**

#### 17 2.1 Theoretical principles

CFDC's, such as the SPIN, are used for ice nucleation measurements by exposing aerosol particles to controlled temperature and relative humidity (RH) conditions. Controlling the temperature and RH is accomplished by first coating the two parallel plates in the main chamber and evaporation section with a thin (~1-2 mm) layer of ice. The water vapor partial pressure directly adjacent to the ice wall is the saturation vapor pressure over ice at the given ice wall temperature. The two walls are held at different temperatures below 0°C and a laminar air stream

flows between the plates. In this idealized configuration, water vapor and heat diffuse from the 1 2 warmer to the cooler wall, leading to linear profiles of water vapor partial pressure and 3 temperature between the two walls. The exponential dependence of saturation vapor pressure on 4 temperature, according to the Clausius-Clapeyron relation, leads to supersaturated conditions 5 (with respect to ice) between the two walls, with a maximum close to the position of the aerosol lamina (Rogers, 1988; Stetzer et al., 2008). Aerosol particles are constrained within this lamina 6 7 and surrounded by two sheath flows passed along each wall. This restricts the aerosol to a 8 narrow range of temperature and supersaturation at which ice nucleation can take place. An 9 example of the chamber flow and thermodynamic profile is shown in Figure 1.

10 A sufficient temperature gradient between the walls results in the water vapor partial pressure 11 in the aerosol lamina exceeding the saturation vapor pressure over liquid water. In this case 12 droplets, in addition to ice, can nucleate on the aerosol particles. Though droplets can be 13 identified using the SPIN optical particle counter (OPC) (Section 2.2), increasing the size difference between droplets and ice helps in distinguishing the two phases. To shrink or 14 15 eliminate droplets while retaining ice crystals, particles pass through an evaporation section after 16 the main chamber (Figure 2). The walls in the evaporation section of the chamber are isothermal and ice coated so the water vapor partial pressure is equal to the saturation vapor pressure over 17 ice. Droplets are therefore unstable and shrink in a manner akin to the Bergeron-Wegner-18 19 Findeisen process (Rogers, 1988; Pruppacher and Klett 1997). Droplets over a critical size will 20 not evaporate completely. The main chamber conditions that generate droplets over this critical 21 size are termed "droplet breakthrough." These conditions are quantified in experiments described 22 in Section 4 and represent an upper RH limit for ice nucleation experiments (only) if droplets and 23 ice crystals are indistinguishable.

#### 1 2.2 SPIN Chamber Design

2 Figure 3 shows a diagram of the SPIN system, illustrating the refrigeration, air flow control, 3 and water flow control components. The temperatures of the two chamber walls and the 4 evaporation section are controlled using compressor-driven refrigeration systems and heater 5 strips affixed to the walls. The warm wall and evaporation section are cooled using a single-stage (with R404A refrigerant) refrigeration loop, while the cold wall is cooled using a two-stage (with 6 7 R404A first stage refrigerant and R508B second stage refrigerant) refrigeration loop. Ten 8 solenoid valves (four for the warm wall, four for the cold wall, and two for the evaporation 9 section) with proportional-integral-derivative (PID) control are used to regulate refrigeration. Thirty 30W heater strips (twelve on the warm wall, twelve on the cold wall, and six on the 10 11 evaporation section) are used to minimize deviations of temperature from the set point by applying heating via twenty-six independent PID controllers (twelve for each of the warm and 12 13 cold walls and two for the evaporation section). Thermocouples that are inserted into the walls 14 and affixed with thermal epoxy are positioned at sixteen locations on each chamber wall and two 15 locations on the evaporation section to map variability in temperature (Figure 2). The chamber 16 itself is machined from aluminum components and junctions are sealed with rubber gaskets.

A filtered and dried sheath flow along each wall is circulated through the chamber using a pump and mass flow controller (MFC). Sample air is drawn into the system by an additional pump. The incoming sample air is drawn into the sheath flow using a knife-edge inlet similar to the one used in the ZINC (Stetzer et al., 2008), which splits the sheath into two flows that move along each wall. The knife-edge also focuses the particle flow to the center of the chamber, limiting the temperature and supersaturation range experienced by the particles. Figure 2 shows the dimensions of the main chamber and evaporation section.

1 After passing through the main chamber and evaporation section, the air stream flows 2 through a linear depolarization OPC that uses four optical detectors for counting, sizing, and 3 differentiating unactivated aerosol particles, droplets, and ice crystals in the 0.4 - 12 µm size 4 range. Figure 4 shows the optical diagram of the OPC. The side scatter detector is used for 5 particle sizing by total scattering intensity, and the backscatter detectors are used to measure P 6 (parallel to the incident laser light) and S (perpendicular to the incident laser light) polarization 7 for phase discrimination: ice crystals have been shown to depolarize more light than water 8 droplets because of their aspherical morphology, and this change in depolarization signal is used 9 to differentiate the two phases (Liou and Lahore, 1974; Nicolet et al., 2010; Clauss et al., 2013; 10 Nichman et al., 2015). The OPC laser (Osela ILS-640-250-FTH-1.5MM-100uM) is a continuous 11 wave 500 mW 670 nm laser with a top-hat beam profile. One of two sets of backscatter optics 12 has a polarizing beam splitter and measure backscattered light in both P and S polarizations (P1 13 and S1, respectively). The second set of backscatter optics measures only the P polarization (P2). The detection angle of both sets of backscatter optics is centered at 135° and has a half angle of 14 15 20°.

16 LabView software is used for instrument control and data acquisition. The SPIN software program consists of several different loops and sub-programs and allows for significant 17 18 automation during operation. The Control Program starts and stops the other modules, updates 19 the displays, controls the instrument set points, watches the alarms, and otherwise supervises the 20 operation of the entire system. The Data Acquisition Loop acquires data for particle events and 21 does so in buffers of typically 500,000 to 1,000,000 points in each channel, with 200 to 400 ns 22 between each data point. This loop acquires data continuously and passes the data buffers to the Data Processing Loop. The Data Processing Loop examines the data buffers acquired by the 23

Data Acquisition Loop and identifies events within those buffers. It extracts the particle data, saves them to disk, and supplies them to the Control Program for display. If data loads are high and the CPU cannot keep up with this processing, some buffers from the Data Acquisition Loop will be ignored and the duty cycle of the acquisition will drop to less than 100%: for 1 Lpm sample flow, this corresponds to particle counts higher than ~3900 per cc.

6 User control of the various SPIN components, including the compressors, valves, and 7 detector is also performed and automated through the LabView interface. Individual actions, 8 such as toggling valves, as well as sequences, such as icing the chamber walls, are controllable 9 through software. The software also includes functionality to create custom sequences, allowing 10 for the majority of operations (including system and compressor startup, cooling the chamber, 11 icing the walls, and running the activation experiments described in Section 3.1) to be automated 12 for increased experimental reproducibility. In addition to the foreground sequences initiated by 13 the user, background sequences can also be run to monitor instrument performance. With remote 14 access enabled through a virtual network computing server (separate from the LabView 15 software), much of the chamber operation can be performed remotely.

16

#### 17 3. Methodology

#### 18 **3.1 Experimental methods**

Before beginning experiments, the chamber is dried, cooled, and the walls are coated with ice. This is accomplished by first flowing dry nitrogen through the chamber via the sample and sheath flow inlets to remove residual moisture; the flow exiting the chamber outlet is routed through a dew point sensor, so the moisture content of the chamber can be directly measured to ensure the dew point is below -40°C. The compressor system is then activated to cool the

chamber to the icing temperature of -25°C. Before icing, the double distilled deionized 18.2 M $\Omega$ 1 2 Millipore (DDI) water in the reservoir is cooled to  $\sim 2^{\circ}$ C to reduce strain on the refrigeration 3 system during icing and to ensure that the wall temperatures do not exceed 0°C over the course 4 of the icing process. With the water reservoir attached to the two-way water pump, the "icing 5 sequence" is activated in the software. This sequence controls the filling and emptying of the 6 chamber with DDI water to form the ice layers. The "ice dwell counter" in software specifies the 7 amount of time the chamber is filled with water and is typically set to 5 s. During and after the 8 icing sequence it is critical to prevent moist room air from entering the chamber, which can 9 cause non-uniform ice on the chamber walls via the formation of frost. This is accomplished by 10 flowing dry air or nitrogen through the sample and sheath flow inlets while allowing the excess 11 flow pressure to be released into the room upstream of the chamber inlets. The entire filling 12 sequence typically lasts ~5 min. After the ice layer has been formed, the dry nitrogen flow 13 through the chamber is continued to ensure that no frost accumulates in the chamber. Subsequent installation of the detector and activation of the sheath pump allows for assessment of 14 15 background frost counts that may bias the reported INP concentrations. This background 16 concentration (typically a few counts per liter) sets the lower detection limit of INP and must ideally be 

converging the wall temperatures decreases chamber supersaturation, and ramping both walls
 allows for temperature scans at the same supersaturation. In all cases, the OPC reports side
 scatter (sizing) and backscatter (depolarization) spectra to infer size, concentration, and phase of
 counted particles.

5 Frost shed from the chamber walls can grow large enough to be counted by the detector, and this phenomenon is more frequent at higher supersaturations. If the number of frost counts is 6 7 comparable to the number of real counts (i.e., on order with the concentration of INPs), the data 8 collected will include a significant artifact. Therefore, periodically measuring the background 9 frost counts with no particles in the chamber (by setting the inlet valve to the filter position) is an 10 important procedure during activation experiments (described below). For converging and 11 diverging wall temperature ramps, this check is performed at the beginning and end of each 12 ramp. For constant supersaturation experiments, this check is performed at fixed time intervals, 13 typically twice per hour. Experiments are automatable using sequences in the SPIN software. 14 These sequences automate the periodic background checks as well as controlling the wall 15 temperature set points. The background concentration increases over time as vapor is transferred 16 from the warm wall to the cold wall, leading to irregularities in the ice layers: as a result, the 17 experiment must be ended once it no longer meets the required levels. The exact time this occurs 18 depends on the particular operating conditions for an experiment but is typically after 2-5 hours 19 of operation.

If the temperature gradient between the warm and cold walls is large (e.g. larger than  $\sim 10$ -15°C, depending on the actual temperatures) the buoyancy of the air adjacent to the warm wall is predicted to overcome the mean flow and causes (upward) flow reversal along the warm wall (Rogers, 1988). The dashed line in the top panel of Figure 5 shows the supersaturation level