# Peer review of "The SPectrometer for Ice Nuclei (SPIN): An instrument to"

_Atmospheric Measurement Techniques, 2015_

## Referee Comment (RC1) · P. DeMott (Referee) · 4 Mar 2016

**General Comment** This paper describes a new ice nucleation device with controlled temperature and humidity conditions in continuous flow. This represents a commercial development based around the continuous flow diffusion chamber concept. A great deal of effort has gone into this characterization, and this paper will likely be of broad interest to the community. I am therefore happy to provide the comments that follow. In my opinion, some key issues for this paper include:

1) This paper is at different times a description of a commercial instrument and at times a record of first developments and studies done by a devoted group of scientists to make this instrument more fully useful. It is difficult to tell where the former ends and the latter begins. Complicating this is a small amount of apparent advertisement that

enters the discussion and conclusions. Some of this is inappropriate in my opinion. I will point out where in specific comments below. What would be helpful is to be a little more careful to state what the SPIN exactly does versus what appear to me to be developments by the first users and future developments (e.g., full automation of many operations). The instrument is not delivered with all of these capabilities, to my knowledge. Can some statement be made upfront that this paper will include a standard description and describe methods developed by this group to utilize the instrument in more advanced ways?

2) Promotion of "AgI" as a calibration ice nucleant, and the (to me) surprising correspondence with literature data, deserves closer inspection. I spent many years in the engineering and generation of "AgI" ice nucleants. One point of studies prior to my time, in the 1960's, was that AgI possesses some of its active characteristics due to impurities (Corrin et al., 1963; 1967). Hence my use of parentheses above. It is certain that the purity of samples sometimes led to vastly different ice nucleation results in laboratory cloud chamber studies. As a minimum, the way that AgI is generated in this study, its size mode (size dependence is clear in much past literature), and some assurance that this was directly comparable to previous studies is needed.

3) A number of details are suggested for addition with regard to the design and construction of SPIN, since this is the first technical description of the device, coming at a time when there is now a substantial literature record of investigations with these kinds of instruments.

4) I found the validation results using AgI versus RH, as well as the homogeneous freezing results not to take the more ideal form I would expect. Hence, some additional discussion and even a new plot (for homogeneous freezing results) may be needed, along with explanation for why these do not look like (my inference) results shown in Richardson et al. (2010) or DeMott et al. (2009), where direct comparison to predictions of homogeneous freezing are made.

5) Finally, there are many parallels of the current findings regarding reverse flow in the work of Richardson (2009). While that was a dissertation and not a peer-reviewed publication, it is available through Colorado State University libraries and was delivered to some of the authors previously. The figure showing a line denoting the reverse flow regime needs to state the flow rate and pressure for which calculations are made. Other issues brought up in the Richardson studies may be present in the results here in the homogeneous freezing regime. It was found at that time that slowing the flow by half was needed to observe homogeneous freezing ideally. The SPIN may not have this limitation, due to the use of the depolarization detector, but I would be surprised if there is not some issue in differentiation when the ice crystals are only growing to 1 micron sizes amongst a field of liquid particles in the time available.

Some specific questions/comments for potentially addressing are listed below. On balance, these could require a number of mostly minor revisions, excepting the request for an additional figure.

**Specific Comments**

*Introduction*

Page 3, Line 7 – The Koop et al. reference is completely about homogeneous freezing. It goes with first part of sentence, but not last, which refers to heterogeneous nucleation.

Page 4, line 2 - Are you certain that lower chamber weight has been achieved in parallel plate designs? My impression is that the SPIN instruments could be heavier than most CFDC's. Can you state a total weight?

Page 4, line 10: I suggest adding Jones et al. (2011) to the list. This CFDC development was actually the first to mimic the original device designed by Rogers (1988), followed later by Saito et al.

*Instrument theory and design*

Page 4. line 21 - Have you measured this ice thickness as 1-2 mm? We have estimated and reported 1/10 of that thickness in the CSU CFDC's based on liquid meltwater measurements. If truly this thick, how much of the inter-plate distance is filled with ice, and do your calculations account for this? It would thus be useful to also report how wide the plate separation is in the SPIN (here, not only in the Figure caption). A horizontal distance scale on the bottom axis of Figure 1 would be useful, and I would presume that such thick ice layers should be shown (and again, accounted for in the calculations).

Page 5, line 16 – How is the refrigeration controlled in the evaporation section of the chamber? Figure 3 only seems to show cooling each wall continuously at one temperature. What constant temperature are the evaporation section walls held at? That is, at the warm or cold wall temperature?

Page 6, line 17 - How is sheath air drying achieved? Can you comment on whether or not a one-way flow of dry air could replace the re-circulated sheath from in the system shown in Fig. 3? CFDC's have been known to be operated with one-way flow of with recirculation. Are both practically possible with the SPIN? Just asking as a scientific point, not with regard to any perceived deficiency.

Page 6, line 22: The knife edge simply places the sample flow initially in the center of the chamber. The limited range of temperature and supersaturation then depends on maintenance of laminar flow conditions, correct?

Page 6, line 23 – It might be useful to point out that the 4:1 growth to evaporation ratio here is much greater than the 2:1 ratio used by Rogers et al. (2001). But this does bring to mind that you should state all of the dimensions in writing here. Especially, it is important to know the cross-sectional dimension of the actual interior chamber region. It is shown as 3 cm in Fig. 2, but the plate separation is 1 cm correct? So with the ice added, the actual separation is 8mm?

Page 6-7: Not addressed anywhere in this section is how walls have been treated to re-

tain ice. I realize that this is found in publications written about chambers that the SPIN design followed, but it is another small detail that deserves discussion somewhere, as stability of ice surfaces is one of the critical issues for these types of instruments.

Page 7 – Is the OPC quite similar to the DMT CASPOL (Glen and Brooks, 2013) or is it very different?

Page 8, line 5 – Concerning the saturation of counting at 3900 per cc for a 1 LPM flow rate, is there any means to correct for "live time" as done for some multi-channel analyzers?

Page 8, lines 9-15 – The question here is what is present as actual functionality with a SPIN versus what is possible via a user's initiative, and has been demonstrated? . That is, should this say "potentially allowing..." and "...much of the chamber operation could conceivably be performed remotely"? The impression is given that full remote operation is already possible, but that is not demonstrated in this paper. These things are all possible for any group with some initiative and technical skills (and some have done so), so without thorough demonstration in this paper, I think that such statements should be avoided.

*Methodology*

Page 8, line 23 – Can you say what type of dewpoint sensor is used, its accuracy, and how well can it resolve a dewpoint temperature of -40 °C? This typically requires an advanced sensor.

Page 9, line 1 – When you say the chamber is cooled to icing temperature, you mean both walls and both wall sections (growth and evaporation) cooled uniformly I assume?

Page 9 – This might be another place to mention how the walls are treated to be "wettable," and what they are made of, materially.

Page 9, lines 6-7 – Is this "dwell counter" time the total time of water in the chamber or the total time that that water remains prior to it being quickly removed? In other words,

how long is water actually in the chamber? The time mentioned is quite short for other CFDC's, yet the ice thickness mentioned is much thicker, so just looking for details that users may wish to know.

Page 9, lines 15-17: "This background concentration (typically a few counts per liter) sets the lower detection limit of INP and must ideally be < 0.1

Page 10, line 5: Doesn't this describe the same effect described above as the "background"? It is not clear. Also, please add that this is uniformly more frequent at higher supersaturations "for the SPIN." Such a generalization may not apply to all CFDCs for this process that is still rather poorly understood, and may vary depending on the method used to obtain wall wettability.

Page 10, lines 6-8 – The statement that frost background of the same order as INP number concentrations means that the INP data will include a significant artifact is understood, but this is likely quantifiable. It certainly does not require that the frost be only 0.1

Page 10, lines 11-12 – It is useful and could be important to report what a typical ramp rate is in terms of cooling rate or d(RH)/dt.

Page 10, line 13: How long are the typical filtered air periods in order to provide a sufficient measure of background frost numbers?

Page 11, lines 13-15 - This seems a variation in the Rogers approach, which may be worth noting. Rogers did not consider along-wall differences, just average wall temperatures in locating the lamina and the flow profile. I am not sure how one uses this information, since the lamina position and velocity profile calculation usually requires assumption of a single value along the walls. What is actually reported then to associate with an INP measurement (for sample conditions and residence time)? Richardson (2009) did consider along wall temperature differences, but he needed FLUENT simulations to resolve the importance of this factor in defining and conditions

and affecting freezing.

Page 13 and Figure 6 – While much of this comment is editorial, I place it here because I feel that this figure requires better explanation. Can the scales and labels be made larger? Please explain the "Size" units or otherwise please explicitly remind of the form of the size parameter. If the standard definition, I could not get the magnitudes correct, or else the wavelength is different than listed for the laser. Also, if the lower size threshold is 0.4 micron diameter, these figures really show particles growing to two orders of magnitude larger in size within the SPIN? Perhaps better, why not show this figure with actual geometric particle size? Finally, has this exercise been done for specific size ranges of INPs, and what are the RH conditions associated with part b (should be stated)?

Page 14, Figure 7 – This figure may bear more discussing as well. It may be helpful here to reiterate that the evaporation region reduces "droplet" fractions. Otherwise, one may wonder why full droplet activation, or anything close to it, is not achieved in this experiment at higher supersaturation, One wonders anyway, since once droplet breakthrough occurs, it seems unusual not to see all droplets retaining sizes well above the OPC lower threshold size (e.g., DeMott et al. 2015). Is this feature due somehow to droplet evaporation after exiting the SPIN and entering the detector region (i.e., due to heat transfer)? The supersaturation seems too high to think that the AgI is not fully activating as droplets at some point. On another matter, when you say validation accuracies, what is the basis for validation? One also would like to know if there is any particular lower ice number concentration value for which this machine learning procedure is easily used. For example, data are shown at INP number concentrations as low as 1 per liter in Fig. 7. Are these values already corrected for background frost? The method seems to be most reliable for INP number concentrations of several 10s per liter. For INP concentrations in much of the mixed-phase cloud regime, how long would one need to integrate ambient data to reliably use this method if INP number concentration is say even 4 per liter? Would this become problematic when droplet

breakthrough occurs and nucleated ice concentrations are quite low? Any statements on the useful range of this otherwise elegant method would be appreciated. Finally, is it possible to speak about the use of a constant particle size that could distinguish ice versus aerosol and drops, perhaps as a function of temperature and RH? Just looking at the data in Figures 6 and 7, it seems possible to use this information for many instruments with similar residence times.

Page 14: How likely is it that AgI is in exactly the same form and size as for other studies referenced in the following results? It surprises me, considering past cloud seeding related literature regarding AgI activation, and how even minor contamination can lead to disparate results.

*Results and comparisons to literature*

Page 14, line 19 – Is this validation or calibration?

Page 15, lines 4-6 – The use of ammonium sulfate here is a classic calibration study for existing CFDCs, but it does lead me to ask about the expectation that sulfate will act only for homogeneous freezing in the lower temperature regime. Heterogeneous ice nucleation activity has been reported for crystalline ammonium sulfate aerosols (Abbatt et al., 2006). It might be important to state why this is not expected in this case. Also, for size selection, can you say the proportion of multiply charged particles? This could affect the homogeneous freezing transition, which I suggest next requires more detailed explanation and discussion.

Page 15, lines 8-9 – The broad sweep of Figure 9, which is nice as a first inspection based on a large suite of measurements, nevertheless suggests to me that the instrument is not reproducing conditions of homogeneous freezing as clearly as observed by some other CFDCs. This transition and the definitive statements made in this section bear some quantitative inspection. What would a line drawn vertically in the figure (e.g., a single experiment) indicate for the active fraction of particles as a function of RH? How would it compare to predictions based on water activity theory

for freezing? This is a standard on which to compare, such as shown in DeMott et al. (2009), Richardson et al. (2010) and some other past literature. Stating clear agreement with Koop et al. (2000) suggests the need for such a presentation. Furthermore, does the figure show false detection of ice at temperatures warmer than homogeneous freezing?

Page 15, lines 15-21 - I must say that it is not sufficient simply to say that the homogeneous freezing behavior is "captured" by the SPIN, versus requiring much more detailed inspection in future studies. As I said above, it does not look ideal (activated fractions do not increase orders of magnitude as shown in other CFDC studies of homogeneous freezing), but this is difficult to tell in this mapped figure. A detailed comparison will be needed, and this suggests an additional figure. What are flow rates and residence time in the chamber? What is efficiency of ice detection below water saturation (i.e., are ice crystals predicted to grow to sizes that will be easily distinguishable via size or polarization)? Noting the discussion in Richardson (2009) and Richardson et al. (2010), especially the need to reduce flow rates in shorter-length CFDCs such as SPIN in order to detect (at least by size) the presence of ice crystals growing above about 1 micron at temperatures below -40C, I wonder if this complication is present in any of the data shown here (i.e., explaining the need for high supersaturations to realize high activated fraction)?

Page 16, line 3 – Editorially, I suggest removing "significantly" here. A few percent above water saturation is not much in order to detect immersion freezing (DeMott et al. 2015 and references therein). It seems that the depolarization detector will be required in many cases. Hence my earlier question about the lower level of sensitivity for ice amongst a field of many liquid particles using this method. It would seem useful to design the instrument with a longer evaporation region.

Page 16, line 7 – AgI aerosols only provide a benchmark if they are carefully produced in exactly the same manner based on a great deal of past literature. Is agreement expected or fortuitous?

Page 16, lines 16-20 – I think that this statement should be omitted. It is not true. Other portable INP chambers have simply switched to two-stage refrigeration systems to achieve lower temperature conditions in the past. See, for example, DeMott et al. (2003) (already referenced here) and Prenni et al. (2007), ground-based and aircraft examples of using a refrigeration system to extend to lower temperatures. It really is not a difficult change for instruments that can be flexible with the configurations used.

*Conclusions*

Page 18, lines 2 to 5 – Here I suggest referring to agreement of your results with Richardson et al. (2009), who quantified uncertainties in a similar manner.

Page 18, lines 7 to 9 – Considering comments above, I consider this a point that is as yet unproven.

Page 18, line 14 – demonstrate, not "validate", in my opinion.

Page 18, lines 16 to 19 – This is a judgement statement that I consider inappropriate for a scientific article. The introduction of this instrument commercially is an important milestone. Nevertheless, the lowered barrier involves a financial investment that is about 4-5 times the cost of a CCN instrument. An individual building an instrument could do so cheaper, and in the process obtain the invaluable experience and investment that my colleagues here have demonstrated is needed to understand and effectively interpret such measurements. It is certainly true that the availability of a commercial instrument could "potentially" lead to an increase in temporal and spatial coverage of INP measurements. This remains to be seen.

Page 18, lines 21 to 23 – The full performance of the depolarization detector would seem to be worthy of additional study. A comparison of ice detection versus other instruments and a standard such as a cloud chamber simulation would be useful for the future.

Page 19, paragraph 1 - The discussion here should reflect any responses to my questions and comments above. Again, utility has been demonstrated, not validated quantitatively.

Page 19, lines 15 to 17 – I suggest striking the statement regarding "SPIN's availability. . ." in preference to something to the effect that "The commercial availability of such a device may allow for increased coverage of INP measurements that will help constrain. . ." In preference to some of this material overall, I wonder if you could say what future needs for demonstration and validation remain (I would say that quantitative intercomparison is needed), and what challenges may remain?

Figure 5 - By "degree of flow reversal" do you mean the fraction of the flow that is reversed? You need to state a total flow and pressure for these calculations right? What would it be at 300 mb, or something more typical of cirrus sampling conditions? Also, it could be useful to superimpose the lamina position on these flow figures.

**Editorial notes**

Page 3, Lines 15-19 – I understand the flow here in saying that CFDC's have been used for both laboratory and field studies, but to be clear, such instruments were designed not to simply detect the conditions for ice nucleation in the laboratory, but to quantify the number concentrations active under a variety of conditions far beyond onset conditions or for specific nucleants. And they were designed for field measurements specifically, from the start. So the statements here are not historically accurate. I suggest, "Several types of instruments have been developed to measure the efficiency of heterogeneous nucleation of cloud droplets and ice crystals. Many of these have applicability for measurements in the laboratory, as well as intended application for field observations."

Page 9, lines 21-22 – Just to advise that use of the term "time dependence" could be confusing, considering stochastic freezing time dependence that is not easily assessed with this type of instrument, and is not what is being referred to here. How about "temporal changes" of INP concentrations?

Page 11. Line 17 – The Kulkarni and Kok reference seems incomplete. Is it a report? Is it freely available?

Page 17, lines 9-10: Are the top 13 thermocouple temperatures then averaged for reported SPIN conditions?

Figure 1 - Re: see above comment on plate separation, in that it could be useful to add a horizontal distance scale under temperature.

Figure 9 – can the typical number concentrations of ammonium sulfate be stated in the figure caption?

**References**

Abbatt, J. P. D., S. Benz, D. J. Cziczo, Z. Kanji, U. Lohmann, O. Möhler, Solid Ammonium Sulfate Aerosols as Ice Nuclei: A Pathway for Cirrus Cloud Formation, Science 313, 1770 (2006).

Corrin, M. L., H. W. Edwards, and J. A. Nelson, 1967: The surface chemistry of condensation nuclei: III. The preparation of silver iodide free of hygroscopic impurities and its interaction with water vapor. J. Atmos. Sci., 21, 565-567.

Corrin, M. L., S. P. Moulik, and B. Cooley, 1967: The surface chemistry of condensation nuclei: III. The adsorption of water vapor on "doped" silver iodide. J. Atmos. Sci., 24, 530-532.

DeMott, P. J., M. D. Petters, A. J. Prenni, C. M. Carrico, S. M. Kreidenweis, J. L. Collett, Jr., and H. Moosmüller, 2009: Ice nucleation behavior of biomass combustion particles at cirrus temperatures, J. Geophys. Res., 114, D16205, doi:10.1029/2009JD012036.

Glen, A., and S. D. Brooks, 2013: A new method for measuring optical scattering properties of atmospherically relevant dusts using the Cloud and Aerosol Spectrometer with Polarization (CASPOL), Atmos. Chem. Phys., 13, 1345-1356.

Jones, H., M. Flynn, P. DeMott, and O. Möhler, 2011: Manchester Ice Nucleus Counter

(MINC) measurements from the 2007 International workshop on Comparing Ice nucleation Measuring Systems (ICIS-2007). Atmos. Chem. Phys., 11, 53-65.

Prenni, A.J., P.J. DeMott, C. Twohy, D.C. Rogers, S.D. Brooks, S.M. Kreidenweis, A.J. Heymsfield and M.R. Poellot, 2007: Examinations of ice formation processes in Florida cumuli using ice nuclei measurements of anvil ice crystal particle residues. J. Geophys. Res., 112, D10221, doi:10.1029/2006JD007549.

Richardson, M., 2009: Making real time measurements of ice nuclei concentrations at upper tropospheric temperatures: Extending the capabilities of the continuous flow diffusion chamber, Dissertation thesis, 268 pp, Colorado State Univ., Fort Collins.

Richardson, M. S., P. J. DeMott, S. M. Kreidenweis, M. D. Petters, M. D., and C. M. Carrico, 2010: Observations of ice nucleation by ambient aerosol in the homogeneous freezing regime. Geophys. Res. Lett., 37, L04806, doi:10.1029/2009GL041912.

---

## Referee Comment (RC2) · Anonymous Referee #2 · 9 Mar 2016

**Review of,** *The SPectrometer for Ice Nuclei (SPIN): An instrument to investigate ice nucleation*

In the submitted manuscript Garimella et al., describe the commercially available SPIN instrument for measuring ice nucleation. To my knowledge the instrument has been on the commercial market for a few years but to date no description of the instrument and its function and capabilities has been available in peer-reviewed scientific literature. While I applaud the large group of co-authors for undertaking the effort of informing the community about this instrument, I do feel that some effort should be made to improve upon the clarity and content of the manuscript before it is eligible for full publication.

The comments of P. DeMott have raised many important issues and at times I reiterate his points for emphasis. As a general comment I do think that if this paper is to be the community reference for SPIN there are many areas where detail must be added. Furthermore, it should be made clear from the abstract onwards what are the 'out-of-the-box' capabilities of the instrument versus contributions that directly result from scientific work and testing. For example, is the machine learning analysis of detector data something that is in part or wholly developed at the user level, or is it an analysis package delivered with the instrument? Given the long list of authors I have no doubt that the current SPIN owners/users have a relatively strong connection, but unless this will form the basis of an organized user group it is not clear that any group acquiring SPIN would have access to and/or benefit from the same community knowledge.

Below I try to summarize items and issues from the general to the specific in the order they arise within the manuscript.
**Itemized points:**

page 3, line 2-5: Aerosol particles are not **required** for nucleation, rather because they are ubiquitous they assist nucleation. In a system with few enough particles homogeneous nucleation could proceed. The way in which particles facilitate the freezing is by changing the free energy barrier to phase transformation. Classical nucleation theory uses bulk thermodynamics to describe the free energy change as a surface/volume competition that is modified by a kinetic pre-factor. The described "mechanisms" are simply phenomenological descriptions of how this proceeds. Thus some thought should be put into accurately phrasing the first two sentences.

page 3, line 7: The Koop et al. (2000) reference is used throughout the manuscript and also within Figs. 5, 9-10. It should be clear that the Koop theory and curve refers to homogeneous freezing in liquid solutions. Thus, the applicability of that theory to freezing in the atmosphere is not always clear – in their manuscript Koop et al. (2000) ignore geometry and consider only large droplets. A very different curve represents homogeneous freezing from the vapor phase – for examples see Fig. 5 in Murray and Jensen (2010) and Fig. 2 in Thomson et al. (2015). See also the discussion of the relevant figures below.

page 3, line 8-12: Although the IPCC reports are highly valuable and useful references, it seems as though citing the recent reports 2x in the papers primary motivating paragraph (Boucher et al., 2013; Stocker et al., 2014) may ignore primary source literature. Perhaps the points being made by the authors would be better emphasized or bolstered if they were slightly expanded and also referred to primary source material, where INP, mixed phase clouds, and/or cloud microphysics are concerned.

page 3, line 21: "RH's" is introduced here but not defined until later on page 4. The abbreviation should be defined at first use and used thereafter.

page 4, line 1-4: The sentence, "Development of parallel plate chamber geometry has

simplified several technical aspects of the chamber design (e.g., lower chamber weight, less complex machining, and simpler refrigeration plumbing than for the cylindrical geometry) at the expense of edge effects (i.e., deviations from ideality at the chamber edges).", is confusing and relegates seemingly important details to the parenthetical. I suggest rephrasing, removing the parentheses, and adding detail to construct a more holistic picture of what is meant.

page 4: The references to the ZINC and PINC systems and papers raise the question of, how much of SPIN can be understood by referring to the literature that describes these earlier systems? Are there things that can be taken out of those papers that are valid for SPIN? I think it is important to make explicit what is SPIN specific versus universal for parallel plate (vertical) CFDCs and/or CFDCs generally.

page 4, §"Theoretical principles": This section reads more like operational principles/procedures. In its current form it would be better suited to come after the design description. If the preference is to keep it in its place then it needs to be re-written in a more general format. However, I would suggest that the authors both outline the general operating principle and describe a suggested user protocol for typical experiments – not necessarily in the same section.

Throughout this section and the following two the manuscript lacks a clear trajectory and often uses imprecise language. Some effort should be made to more clearly delineate between the theoretical working principle, what *could possibly be* done in SPIN given the intended engineering, and what in fact *is* done in (with) SPIN.

For example,

page 4, line 19-20: "Controlling the temperature and RH is accomplished ...." It seems to me that temperature control has nothing to do with icing the chamber walls. The temperature is controlled by refrigeration and heaters, the saturation condition is established by icing the walls and controlling the temperature gradients. Likewise (line 23), "The two walls are held at different temperatures ...." That is a statement of operational protocol, because within some range it appears you could set the temperatures to whatever you want, they could even be equal. It would be better to say something like, 'In order to establish the necessary vapor supersaturations for nucleation the ice coated chamber walls are held at different temperatures ...'

page 5, line 6: How well are the particles constrained within the sample flow lamina? Garimella had a presentation at the 2015 AGU Fall meeting suggesting that this is a source of uncertainty (Garimella et al., 2015). That presentation or a manuscript in preparation could be referred to, to say something general about this.

page 5, line 12-end: The effectiveness of the evaporation section is largely dependent on the relevant residence times, perhaps this could be addressed? Isn't the critical droplet size related to the time a droplet might have to evaporate? Is it important droplets completely evaporate? Or can they simply decrease in size to below a threshold?

page 7, lines 6-10: Both the aspherical morphology and depoloraization associated with ice crystals come from the anisotropy of the ice as a material. Ice crystals have fast and slow growth directions (that lead to their macroscopic morphology, e.g. Cahoon et al., 2006; Wettlaufer et al., 1999) and are also optically birefringent (e.g., Thomson et al., 2009; Lekner, 1991, 1999, and references therein). Both aspects are important.

page 7, line 16: "LabView software is used for instrument control and data acquisition. The SPIN software program consists of several different loops and sub-programs and allows for significant automation during operation. The Control Program starts and stops the other modules, updates the displays, controls the instrument set points, watches the alarms, and otherwise supervises the operation of the entire system." Given the lack of specificity much of this text seems superfluous, it would suffice to say: The SPIN instrument is operated and controlled via a LabView master control program. (or alternatively add pertinent information)

page 8, line 5: What happens when particle counts approach this level? What is

ignored? Does this bias results? Is it recommended that a dilution flow is added to keep raw particle counts below this threshold?

page 8, lines 6-15: Is the entire icing scheme fully automated as one could believe from this paragraph?

page 9, line 4: "icing sequence" Quotation marks are generally used to set off material that represents quoted or spoken language. At times they are used to show sarcasm. Here neither appears to be the case – they seem to be indicating LabView program components. If these terms/phrases are being emphasized it would be better to use a different type-setting tool (bold, italics, etc.). The same mistake is made throughout the manuscript, eg., "ice dwell counter," etc.

page 9, line 23 - page 10 line 2: "For the former, diverging the wall temperatures increases the chamber supersaturation,... and ramping both walls..." The uses of 'diverging' and 'converging' are awkward in this sentence, I suggest rephrasing (for example, 'increasing the temperature gradient between the walls'). They are correctly used a bit further in the text. Also, the walls are not 'ramped'. The temperatures are increased and decreased or temperatures are ramped.

page 11, line 12: extra comma after "Rogers"

§3.2 Data processing and methods: Given the audience of AMT this section lacks significant detail and some concentrated effort needs to go into a major re-writing. While machine learning and other data analysis techniques may be known to some, I do not think one can assume they are part of the atmospheric measurement community vocabulary. Detail and references must be added to the section, and if deemed excessive for direct inclusion I suggest that fundamental information now not included could become part of a supplement. For example in Figure 6 data from a GMM-KDE is presented – could a supplement include a step-by-step explanation of the analysis process, perhaps including some idealized system (where the separation is more clearly bi-modal)? These techniques seem to be a powerful tool, but it is hard to appreciate in

the limit of the single example and poor explanation.

page 12, line 1: "supervised machine learning" is introduced without a reference. Include a reference so that those unfamiliar with this have a standard text to which to refer.

page 12, lines 3-4: "The also require fewer assumptions to be ...." Machine learning requires fewer assumptions than what?

page 12, line 6: I suggest rephrasing to say, "... historically been analyzed using *post-evaporation section* particle size as...."

page 12, line 10: I suggest rephrasing to say, "... than the ice size and that droplets *above that size* do not survive..."

page 12, line 15: By "efficiently" do you mean completely?

page 13, line 4: The introduction of Kernel Density Estimation should include a reference.

page 13, line 7: The introduction of the 4-d Gaussian mixture model (GMM) should include a reference and likely Mixture and Model should begin with capital letters.

page 13, line 10: "aerosol only" Again a misuse of quotation marks. However, this time the phrase is also left undefined. If this is to be used to indicate a defined procedure/measurement it should be explicitly defined. The same is true of "aerosol + ice" and "ice only."

page 13, line 18: Gaussian kernel support vector machine needs a reference.

page 13, lines 20 –: The final sentence of the paragraph, "Since a condensation ... " is long, awkwardly appended and does not follow from the previous material. Make this an independent paragraph that clearly explains how activated fractions are calculated.

page 14, lines 4-6: "3-class supervised machine learning (bootstrap aggregated decision trees....)" At least two missing references here, 3-class supervised machine learning and bootstrap aggregated decision trees? Also missing is any explanation of why these methods are used versus what is described before and how are they different.

As a general comment to supplement the specific comments regarding the data processing section. Given this is not a computer science journal then one cannot expect that everyone is well versed in machine learning, KDEs, GMMs, etc. As a reader it is very disturbing that much of this seems to be introduced as little more than buzz-words with acronyms. Some effort needs to be made to indicate what in practice is being done at each step. As I understand machine learning is used to predict outcomes – what exactly are the outcomes being predicted? How are the training vectors selected? Are KDEs used to make PDFs of size or of all OPC variables, it is not clear? A 2D GMM-KDE is used for visualization but a 4D used for the actual calculation? Does that mean that the 4D does a better job than we can understand by looking at the figures? Otherwise, what is the difference?

Finally, should the community expect that all SPIN data should be reported in this manner (i.e. As a result of this type of analysis?)? Is this a computationally time consuming process, or is it quick and completely automated? Can (is) such an analysis be implemented in real time, such that one observes the results while running an experiment? What part of the analysis (training data, etc.) needs to be implemented separately for individual detectors on different units, or are all detectors identical to within the uncertainty?

page 18, line 14: What is meant by "validate"? Validate the performance as what? Perhaps this is too strong?

page 19, line 2: Here given what has been presented within the manuscript I take "homogeneous INP" to refer to homogeneous freezing of solution droplets. It is important to make the distinction between homogeneous freezing from solution and homogeneous freezing from the vapor.

page 23, line 20: Kulkarni et al. ref. missing doi (doi missing from many refs)

page 23, line 21: Kulkarni and Kok reference seems incomplete. Is this a report? In a series? Is this document available publicly?

page 26, line 4: The Stocker IPCC reference is missing co-authors or an et. al.

page 31, Figure 5: What are the units of the color bar? "The color scale shows the degree of flow reversal..." What is meant by "degree"? The small subplots can be better connected to the intensity plot. I assume for example in the flow reversal subplot -30 represents the warm wall and -45 the cold wall? This could be indicated in the figure and/or caption. Also, I take the color intensity to represent the maximum flow reversal velocity – is that true? Or is it a mean flow reversal velocity? Also it must be made explicit that the Koop curve represents the onset of homogeneous freezing of liquid solutions (droplets). Given the dashed flow reversal line the phase space that can be explored without flow reversal within the chamber is quite limited. Is this an issue, does it affect the utility of this (and similar) instruments? It would be beneficial to discuss this issue in the text where the flow reversal and figure appear.

Finally, is the plan for this figure to appear in a single column within the 2 column journal? If so the subplots and legend will become very difficult to make out. Perhaps the figure should be redrawn to optimize it for the 1 (or 2) column size?

page 32, Figure 6: Again think about the figure size in 1 or 2 columns. The panels will be very small, and it is difficult to read any of the numbers/axes labels and legends in the current full page format. I had to zoom into the *.pdf many times to recognize that the support vector points were plotted as open circles enclosing data points.

What are the $\log_{10}$(size) units? I have not been able to convince myself of the meaning of these axes. In the upper 3 panels time 2 - time 1 does not look like 0 as it appears in (c). This could be due to rescaling because the intensity shading in (a)-(c) changes. I suggest that an absolute color scheme is chosen (perhaps including a wider color

spectrum), such that colors in all panels represent the same probability density.

page 33, Figure 7: Again think about the figure size in 1 or 2 columns.

What does classification accuracy of 99% mean? It seems that some points would be much more certain than others. Does this mean that the total number of each is identified to a 99% certainty level? In all 3 dimensions? The 3D plot shows an area where the 3 colors (particles, ice, droplets) merge. I would assume in this area uncertainty would be high, while far from this uncertainty would be small? In the right-hand panels why does ice fraction equal water fraction at conditions sub-saturated with respect to water? Is this an area of phase space where the droplets and ice cannot be distinguished beyond the level of uncertainty? If so is this a result of small particle sizes? Is there a critical ice (droplet) size to distinguish solid from liquid?

page 34, Figure 8: These panels are probably better arranged vertically for the 2 column format. Is the sigmoid fit used as a scaling factor for results? Is sigmoid the appropriate fit – perhaps the efficiency is unity to a size cutoff, below which the relationship is linear? Without one or more further data points the choice of fit should be supported based on some reasoning. Was the previously presented data (Figs. 6,7) processed using these relationships? I would suggest discussing these issues in the text where Figure 8 appears.

page 35, Figure 9: Increase the size of the legend and labels. Also, again specify that the "expected homogeneous freezing" refers to aqueous phase homogeneous freezing. See my previous references to the Koop article. In such a plot it is difficult to also incorporate uncertainty, but perhaps some indication of the uncertainty range of activated fraction can be given?

page 36, Figure 10: See early comments with respect to formatting, font etc.

page 37, Figure 11: See early comments with respect to formatting, font etc. I wonder if this figure could also be incorporated into the earlier discussion of flow reversal etc.
Comparing with Figure 5 and given that $S_{ice}$=1.3, then flow reversal should be present in (e). However, I see no evidence of this and the scale incorporates only positive velocities. The two plots are not self-consistent, please explain.

Let me reiterate where I see a need for major revisions.

• The data processing section is difficult to understand and incomplete in its current form.
• Operating procedures and analysis etc. specific to SPIN should be clearly delineated from general discussions of the CFDC principle (eg., How the saturations gradient is established.). Furthermore, I encourage the authors to highlight what they consider to be the **best use practices** of SPIN given the current state of understanding. A simple example would be the utilization of an icing temperature of -25°C and time of 5 s – were these observed to somehow lead to optimal experimental conditions, or simply chosen at random?
• Clearly delineate what part of this manuscript represents out-of-the-box SPIN operation/ measurements versus user implemented protocols and analysis. For example, is the water reservoir cooling to 2°C standard for SPIN?
• If necessary re-write the abstract to reflect the scope of the standard versus user enabled capacity of SPIN.

**References**

Boucher, O., Randall, D., Artaxo, P., Bretherton, C., Feingold, G., Forster, P., Kerminen, V.-M., Kondo, Y., Liao, H., Lohmann, U., et al.: Clouds and aerosols, in: Climate change 2013: The physical science basis. Contribution of working group I to the fifth assessment report of the intergovernmental panel on climate change, pp. 571–657, Cambridge University Press, 2013.
Cahoon, A., Maruyama, M., and Wettlaufer, J. S.: Growth-Melt Asymmetry in Crystals and Twelve-Sided Snowflakes, Physical Review Letters, 96, 2006.

[Figure]

Garimella, S., Voigtländer, J., Kulkarni, G., Stratmann, F., and Cziczo, D. J.: Biases in field measurements of ice nuclei concentrations, AGU Fall Meeting, 2015.

Koop, T., Luo, B. P., Tsias, A., and Peter, T.: Water activity as the determinant for homogeneous ice nucleation in aqueous solutions, Nature, 406, 611–614, 2000.

Lekner, J.: Reflection and refraction by uniaxial crystals, Journal of Physics: Condensed Matter, 3, 6121–6133, 1991.

Lekner, J.: Reflection by uniaxial crystals: polarizing angle and Brewster angle, Journal of the Optical Society of America A, 16, 1999.

Murray, B. J. and Jensen, E. J.: Homogeneous nucleation of amorphous solid water particles in the upper mesosphere, Journal of Atmospheric and Solar-Terrestrial Physics, 72, 51 – 61, doi:http://dx.doi.org/10.1016/j.jastp.2009.10.007, 2010.

Stocker, T. F. et al.: Climate change 2013: the physical science basis: Working Group I contribution to the Fifth assessment report of the Intergovernmental Panel on Climate Change, Cambridge University Press, 2014.

Thomson, E. S., Wilen, L. A., and Wettlaufer, J. S.: Light scattering from an isotropic layer between uniaxial crystals, Journal of Physics: Condensed Matter, 21, 195 407 (10pp), doi:10.1088/0953-8984/21/19/195407, 2009.

Thomson, E. S., Kong, X., Papagiannakopoulos, P., and Pettersson, J. B. C.: Deposition-mode ice nucleation reexamined at temperatures below 200 K, Atmospheric Chemistry and Physics, 15, 1621–1632, doi:10.5194/acp-15-1621-2015, 2015.

Wettlaufer, J. S., Dash, J. G., and Untersteiner, N., eds.: Ice Physics and the Natural Environment, vol. 56 of *NATO ASI Series I*, SPRINGER-VERLAG, Heidelberg, 1999.

---

## Referee Comment (RC3) · Anonymous Referee #2 · 9 Mar 2016

One more point of importance with regard to using the homogeneous freezing line of Koop. Typically a nucleation rate must be specified when plotting this curve. For example, the value $J = 5 \times 10^8$ cm$^{-3}$s$^{-1}$ from Hoose and Möhler (2012) is often used.

**References**

Hoose, C. and Möhler, O.: Heterogeneous ice nucleation on atmospheric aerosols: a review of results from laboratory experiments, Atmos. Chem. Phys., 12, 9817–9854, 10.5194/acp-12-

9817-2012, 2012.

---

## Referee Comment (RC4) · Anonymous Referee #3 · 26 Mar 2016

I do not have many additional comments to add to the two previous thorough reviews as the main points that need clarification are already pointed out. What I would like to see is some preliminary results of ambient measurements (e.g. at -30°C at a defined supersaturation). From the statement on page 19, line 17, I am assuming that it is also meant for field measurements, but from what has been shown in the manuscript, I would tend to think that the SPIN is a laboratory CFDC. The main purpose of doing these measurements is to see if the background counts (through a filter) is significantly below the INP concentrations. As the ice nucleation community already know, INP are scarce in the atmosphere and background/noise coming from something else than actual INP can totally make the instrument unusable for field studies.

Specific comments :

Page 4 – line 21 : how did the authors measure the thickness of the ice layers ? Is the thickness consistent from an icing to another ? Please clarify.

Page 9 – lines 16,17 : Can the authors show a plot of the background concentration they are considering as ideal before an INP measurement ? It would also be good to include one showing when it is not ideal to keep on sampling.

Page 10 – line 18 : The operation time before stopping the measurements is quite broad (2-5 hours). Could the authors clarify this part ? Which specific conditions (supersaturation, temperature) lead to shorter or longer operating time.

Figures Please increase the font of the figures (from figure 6 to 11).

---

## Author Comment (AC1) · 6 May 2016

Please see the attached supplement for the author responses to referee comments, updated manuscript, and tracked changes document.

Please also note the supplement to this comment:
http://www.atmos-meas-tech-discuss.net/amt-2015-400/amt-2015-400-AC1-supplement.zip